# Adaptive Navigation Performance Evaluation Method for Civil Aircraft Navigation Systems with Unknown Time-Varying Sensor Noise

**DOI:** 10.3390/s24165093

**Published:** 2024-08-06

**Authors:** Yuting Dai, Jizhou Lai, Qieqie Zhang, Zhimin Li, Rui Liu

**Affiliations:** College of Automation Engineering, Nanjing University of Aeronautics and Astronautics, No. 29 General Avenue, Nanjing 211106, China; yt_dai@nuaa.edu.cn (Y.D.); zhangqieqie@nuaa.edu.cn (Q.Z.); lizhimin@nuaa.edu.cn (Z.L.); liuruizhx@nuaa.edu.cn (R.L.)

**Keywords:** actual navigation performance (ANP), required navigation performance (RNP), civil aircraft, time-varying unknown noise, position estimation covariance matrix (PECM), variational Bayesian (VB)

## Abstract

During civil aviation flights, the aircraft needs to accurately monitor the real-time navigation capability and determine whether the onboard navigation system performance meets the required navigation performance (RNP). The airborne flight management system (FMS) uses actual navigation performance (ANP) to quantitatively calculate the uncertainty of aircraft position estimation, and its evaluation accuracy is highly dependent on the position estimation covariance matrix (PECM) provided by the airborne integrated navigation system. This paper proposed an adaptive PECM estimation method based on variational Bayes (VB) to solve the problem of ANP misevaluation, which is caused by the traditional simple ANP model failing to accurately estimate PECM under unknown time-varying noise. Combined with the 3D ANP model proposed in this paper, the accuracy of ANP evaluation can be significantly improved. This enhancement contributes to ensured navigation integrity and operational safety during civil flight.

## 1. Introduction

The implementation of performance-based navigation (PBN) of civil aircrafts in recent years has greatly enhanced the capacity of civil airspace. With the development of airborne navigation systems, especially global navigation satellite systems (GNSSs), the implementation of required navigation performance (RNP) with higher performance requirements has become a trend for future civil aviation air traffic management [1,2,3,4].

RNP is an advanced civil aviation navigation procedure that defines the essential capability of the airborne navigation system in the form of a performance threshold tunnel when operating in specified airspace. The flight management system (FMS) requires fault detection, fault isolation, and system reconfiguration capabilities, and it uses hardware redundancy and model-based analytic redundancy methods for accurate fault identification [5,6,7]. The FMS also needs to accurately evaluate a civil aircraft’s navigation capability in real-time to determine if it meets the current RNP threshold. An incorrect evaluation result will cause misjudgement by the crew, which will lead to serious threats to flight safety.

The FMS uses the probability circle radius (95%) of estimated position uncertainty (EPU) to represent the actual navigation performance (ANP) [8]. The existing ANP evaluation method focuses on the improvement of the EPU model [9,10,11]. As shown in Figure 1, the evaluation accuracy of ANP depends on the precision of position uncertainty estimation provided by the FMS.

Airborne integrated navigation systems usually use empirical positioning error models of each navigation sensor for position calculation to ensure computational efficiency [12]. However, the performance of the primary navigation source during civil flight is time-varying and unpredictable [13]. In this case, the use of an empirical error model will lead to inaccurate calculation of the position error covariance matrix (PECM) by the integrated navigation system, which in turn leads to inaccuracy of the ANP value [14].

Accurate estimation of the time-varying PECM is a prerequisite for ANP calculations. The exact solution of the time-varying PECM caused by inaccurate and unknown noise cannot be obtained in-flight, and it is more common to estimate the PECM using approximation methods. There are numerous approximation methods, including the maximum likelihood method, maximum a posterior method, expectation–maximization method, variational Bayesian methods, etc. [15]. Variational Bayesian (VB) methods are a combination of Bayesian reasoning and variational calculus. They can perform approximate posterior inference at a low computational cost by choosing an appropriate conjugate prior distribution [16,17]. However, the existing VB-based estimator can estimate an unknown and slowly varying measurement noise by choosing the appropriate conjugate prior distribution [18]. But when the estimator uses an inaccurate process noise matrix, it will introduce inaccuracy in the PECM estimation; therefore, the tracking capability for time-varying unknown system noise needs to be improved.

This paper aims to solve the ANP misestimation cost due to unknown time-varying sensor noise. An adaptive 3D ANP evaluation method based on variational Bayesian was proposed. The main contributions of this paper are as follows: (1) A novel VB-based PECM estimator was constructed to accurately estimate both the predicted error covariance matrix and the measurement noise matrix in integrated navigation systems. This approach effectively addresses the issue of inaccurate estimation caused by unknown noise in the integrated navigation system input. (2) The fading memory index weighting is used to improve the tracking capability of time-varying noise, and the accuracy of the actual navigation performance of the integrated navigation system is further improved. (3) To address the limitations of the traditional ANP model, a 3D ANP model was developed that was specifically designed to overcome issues such as missing evaluation dimensions and model inaccuracy. Finally, a simulation was performed to prove the algorithm.

The structure of the proposed airborne integrated navigation system and ANP evaluation is shown in Figure 2. The FMS uses the inertial reference system (IRS) as the reference system and the GNSS and land-based radio system as auxiliary systems. The accurate PECM can be obtained using the proposed VB-based estimator, and by applying PECM to the 3D ANP model, the FMS can output a 3D-ANP evaluation result for civil aircrafts during RNP flight.

## 2. Traditional Airborne ANP Evaluation with Kalman Filter

The traditional airborne ANP evaluation is based on the Kalman filter. The discrete-time linear state-space model of the airborne integrated system is as follows:(1)$$\left\{\begin{array}{c} X_{k}=\Phi_{k,k-1}X_{k-1}+W_{k-1}\\ Z_{k}=H_{k}X_{k}+V_{k} \end{array}\right.$$

Gaussian system noise $W_{k-1}\sim N(0,Q_{k-1})$ and Gaussian measurement noise $V_{k}\sim N(0,R_{k})$ are irrelevant. 

The process of integrated navigation based on the Kalman filter is as follows:
(1)Time update:(2)$${\hat{X}}_{k|k-1}=\Phi_{k,k-1}{\hat{X}}_{k-1|k-1}$$(2)Measurement update:(3)$$P_{k|k-1}=\Phi_{k/k-1}P_{k-1}\Phi_{k/k-1}^{T}+Q_{k-1}$$
(4)$$K_{k}=P_{k|k-1}{H_{k}}^{T}{\left(H_{k}P_{k|k-1}{H_{k}}^{T}+R_{k}\right)}^{-1}$$
(5)$$P_{k|k}=(I-K_{k}H_{k})P_{k|k-1}{\left(I-K_{k}H_{k}\right)}^{T}+K_{k}R_{k}K_{k}^{T}$$

$P_{k|k}$ is the estimated error matrix (EEM) of the Kalman filter, which includes the position estimate error (PEE) of the airborne integrated navigation system.$P_{k|k-1}$ is the one-step prediction error matrix. FMS uses the PECM obtained according to $P_{k|k}$ as the ANP calculation input. PECM Λ for the horizontal plane can be expressed as follows:(6)$$\Lambda =A\Lambda_{m}A^{-1}$$
where:(7)$$\Lambda_{m}=\left[\begin{array}{cc} \sigma_{mx}^{2} & 0\\ 0 & \sigma_{my}^{2} \end{array}\right]$$

A is the similarity transformation matrix of Λ. Then, the ANP in the horizontal plane can be expressed as follows:(8)$$\mathrm{ANP}=k\times l_{\mathrm{major}}$$

$l_{\mathrm{major}}=\max[\sigma_{mx},\sigma_{my}]$ is the major axis of the PEE ellipses, and *k* is the scale factor of the 95% probability elliptic integral:(9)$$k=\frac{0.4852}{ratio^{3}}+1.9625$$

The Kalman filter is the optimal recursive linear variance estimation algorithm with accurate $Q_{k}$ and $R_{k}$ values. However, active FMS empirically sets $Q_{k}$ and $R_{k}$ according to the sensor error characteristics. As shown in Figure 3, such customized parameters cannot reflect the changing characteristics of uncertain noise [19]. The use of inaccurate $Q_{k}$ and $R_{k}$ values will result in substantial estimation errors in the PECM, leading to misevaluation of ANP. Therefore, a novel VB-based PECM approximation method will be proposed. 

## 3. Adaptive Navigation Performance Evaluation Method Based on Adaptive PECM Estimation

### 3.1. PECM Approximation Based on VB with Unknown Time-Varying Noise

As shown in Figure 4, to obtain a 3D ANP safety boundary, the PECM of the airborne integrated navigation system with uncertain and time-varying noise needs to be estimated more accurately. The variational Bayesian approximation is used to determine $X_{k}$, $R_{k}$, and $P_{k|k-1}$ to obtain an accurate estimation of $P_{k|k}$. 

$X_{k}$, $R_{k}$, and $P_{k|k-1}$ are independent, and the variational approximation of joint posterior PDF is as follows:(10)$$p(X_{k},P_{k|k-1},R_{k}|Z_{1:k})\approx q(X_{k})q(P_{k|k-1})q(R_{k})$$

The posterior PDF of p(·) is represented within q(·). We use the Kullback–Leibler (KL) divergence to describe the approach degree of the approximate posterior distribution and true posterior distribution.
(11)$$\begin{array}{l} D_{KL}[p(X_{k},P_{k|k-1},R_{k}|Z_{1:k})||q(X_{k})q(P_{k|k-1})q(R_{k})]=\\ \int \int \int q(X_{k})q(P_{k|k-1})q(R_{k})\log\frac{q(X_{k})q(P_{k|k-1})q(R_{k})}{p(X_{k},P_{k|k-1},R_{k}|Z_{1:k})}dX_{k}dP_{k|k-1}dR_{k} \end{array}$$

$q(X_{k}),q(P_{k|k-1}),q(R_{k})$ can be acquired by minimizing the KL divergence $D_{KL}$.
(12)$$\left[q(X_{k}),q(P_{k|k-1}),q(R_{k})]=\arg\min(D_{KL}\right)$$

Taking the derivative of $q(X_{k}),q(P_{k|k-1}),q(R_{k})$ results in the following:(13)$$q(X_{k})\propto \int q(X_{k})\log p(Z_{k},X_{k},P_{k|k-1},R_{k}|Z_{1:k})dX_{k}$$
(14)$$q(P_{k|k-1})\propto \int q(X_{k})\log p(Z_{k},X_{k},P_{k|k-1},R_{k}|Z_{1:k})dP_{k|k-1}$$
(15)$$q(R_{k})\propto \int q(R_{k})\log p(Z_{k},X_{k},P_{k|k-1},R_{k}|Z_{1:k})dR_{k}$$

The above Equations (13) to (15) cannot be directly solved analytically due to mutual coupling. By calculating the expectation of these equations, we can obtain the following:(16)$$\log q(\delta )=E_{\Psi^{-\delta }}[\log p(\Psi ,Z_{1:k})]+C_{\delta }$$

Within $\Psi ≜(X_{k},P_{k|k-1},R_{k})$, E(·) represents the expectation calculation, δ is an arbitrary element of Ψ, $\Psi^{-\delta }$ indicates the set of elements of Ψ except for δ, and $C_{\delta }$ is a constant term associated with δ.

For a linear system, the PDF of $p(X_{k}|Z_{1:k},P_{k|k-1})$, $p(Z_{k}|X_{k},R_{k})$, $P_{k|k-1}$, and $R_{k}$ are Gaussian. N(·;μ,σ) denotes a Gaussian distribution with a mean of μ and a covariance of σ.
(17)$$p(X_{k}|Z_{1:k},P_{k|k-1})=N(X_{k};{\hat{X}}_{k|k-1},P_{k|k-1})$$
(18)$$p(Z_{k}|X_{k},R_{k})=N(Z_{k};H_{k}X_{k},R_{k})$$

In variational Bayesian approximation, the inverse Wishart distribution is usually used as the conjugate distribution of the Gaussian distribution to reduce the iterative computation. Therefore, the prior distribution of $P_{k|k-1}$ and $R_{k}$ is selected as the inverse Wishart distribution to derive the posterior distribution of $X_{k}$. IW(·;θ,ω) denotes an inverse Wishart distribution with degrees of freedom (DOF) θ and scale matrix ω.
(19)$$p(P_{k|k-1}|Z_{1:k-1})=IW(P_{k|k-1};y_{k|k-1},Y_{k|k-1})$$
(20)$$p(R_{k}|Z_{1:k-1})=IW(R_{k};u_{k|k-1},U_{k|k-1})$$

The joint posterior PDF Equation (10) can be rewritten as follows:(21)$$p(X_{k},P_{k|k-1},R_{k}|Z_{1:k})=p(Z_{k}|X_{k},R_{k})p(X_{k}|Z_{1:k},P_{k|k-1})p(P_{k|k-1}|Z_{1:k})p(Z_{1:k})$$

By substituting Equations (19) and (20) into Equation (21), we obtain the following:(22)$$\begin{array}{c} p(X_{k},P_{k|k-1},R_{k}|Z_{1:k})=N(Z_{k};h(X_{k}),R_{k})N(X_{k};{\hat{X}}_{k|k-1},P_{k|k-1})\\ \;\;\;\;\;\;\;\;\;\times IW(P_{k|k-1};y_{k|k-1},Y_{k|k-1})IW(R_{k};u_{k|k-1},U_{k|k-1})p(Z_{1:k-1}) \end{array}$$

The $\log p(\Psi ,Z_{1:k})$ in Equation (16) can be derived as follows:(23)$$\begin{array}{ll} \log p(\Psi ,Z_{1:k}) & =-0.5\log\left|R_{k}\right|-0.5{\left(Z_{k}-H_{k}X_{k}\right)}^{T}R_{k}^{-1}(Z_{k}-H_{k}X_{k})\\ & -0.5\log\left|P_{k|k-1}\right|-0.5{\left(X_{k}-{\hat{X}}_{k|k-1}\right)}^{T}P_{k|k-1}^{-1}(X_{k}-{\hat{X}}_{k|k-1})\\ & -0.5(m+u_{k|k-1}+1)\log\left|R_{k}\right|-tr(U_{k|k-1}R_{k}^{-1})\\ & -0.5(n+y_{k|k-1}+1)\log\left|P_{k|k-1}\right|-tr(Y_{k|k-1}P_{k|k-1}^{-1})+C_{\psi } \end{array}$$
where *n* is the dimension of $X_{k}$, and m is the dimension of $Z_{k}$.

Then, an iterative approximation of $q(X_{k}),q(P_{k|k-1}),q(R_{k})$ can be obtained. Let $\Psi =P_{k|k-1}$, deploying Equations (23) to (16):(24)$$\begin{array}{ll} \log q^{\left(i+1\right)}(P_{k|k-1}) & =-0.5(n+y_{k|k-1}^{\left(i\right)}+2)\log\left|P_{k|k-1}\right|\\ & -tr\left[(Y_{k|k-1}^{\left(i\right)}+M_{k}^{\left(i\right)})P_{k|k-1}^{-1}\right]+C_{P} \end{array}$$
(25)$$M_{k}^{\left(i\right)}=(X_{k}-{\hat{X}}_{k|k-1}^{\left(i\right)}){\left(X_{k}-{\hat{X}}_{k|k-1}^{\left(i\right)}\right)}^{T}$$

After *i* + 1 iterative operations, $q(P_{k|k-1})$ can be updated as follows:(26)$$q^{\left(i+1\right)}(P_{k|k-1})=IW(P_{k|k-1};y_{k|k-1}^{\left(i\right)},Y_{k|k-1}^{\left(i\right)})$$
(27)$$y_{k|k-1}^{\left(i+1\right)}=y_{k|k-1}^{\left(i\right)}+1$$
(28)$$Y_{k|k-1}^{\left(i+1\right)}=Y_{k|k-1}^{\left(i\right)}+M_{k}^{\left(i\right)}$$

Similarly, let $\Psi =R_{k}$:(29)$$\begin{array}{ll} \log q^{\left(i+1\right)}(R_{k}) & =-0.5(m+u_{k|k-1}^{\left(i\right)}+2)\log\left|R_{k}\right|\\ & -tr\left[(U_{k|k-1}^{\left(i\right)}+A_{k}^{\left(i\right)})R_{k}^{-1}\right]+C_{R} \end{array}$$
(30)$$A_{k}^{\left(i\right)}=(Z_{k}-H_{k}X_{k}^{\left(i\right)}){\left(Z_{k}-H_{k}X_{k}^{\left(i\right)}\right)}^{T}$$

The $q(R_{k})$ value can be updated as follows:(31)$$q^{\left(i+1\right)}(R_{k})=IW(R_{k};u_{k}^{\left(i\right)},U_{k}^{\left(i\right)})$$
(32)$$u_{k}^{\left(i+1\right)}=u_{k}^{\left(i\right)}+1$$
(33)$$U_{k}^{\left(i+1\right)}=U_{k}^{\left(i\right)}+A_{k}^{\left(i\right)}$$

Let $\Psi =X_{k}$:(34)$$\begin{array}{ll} \log q^{\left(i+1\right)}(X_{k}) & =-0.5{\left(Z_{k}-h(X_{k}^{\left(i\right)})\right)}^{T}{\left[R_{k}^{-1}\right]}^{\left(i+1\right)}(Z_{k}-h(X_{k}^{\left(i\right)}))\\ & -0.5{\left(X_{k}-{\hat{X}}_{k|k-1}^{\left(i\right)}\right)}^{T}{\left[P_{k|k-1}^{-1}\right]}^{\left(i+1\right)}(X_{k}-{\hat{X}}_{k|k-1}^{\left(i\right)})+C_{X} \end{array}$$
where:(35)$${\left[R_{k}^{-1}\right]}^{\left(i+1\right)}=(u_{k}^{\left(i+1\right)}-m-1){\left[U_{k}^{\left(i+1\right)}\right]}^{-1}$$
(36)$${\left[P_{k|k-1}^{-1}\right]}^{\left(i+1\right)}=(y_{k}^{\left(i+1\right)}-n-1){\left[Y_{k}^{\left(i+1\right)}\right]}^{-1}$$

The $q(X_{k})$ value can be updated as follows:(37)$$\begin{array}{ll} q^{\left(i+1\right)}(X_{k}) & =\frac{1}{C_{X}}p^{\left(i+1\right)}(Z_{k}|X_{k})p^{\left(i+1\right)}(X_{k}|Z_{1:k-1})\\ & =N(X_{k};{\hat{X}}_{k|k}^{\left(i+1\right)},P_{k|k}^{\left(i+1\right)}) \end{array}$$
(38)$$C_{X}=\int p^{\left(i+1\right)}(Z_{k}|X_{k})p^{\left(i+1\right)}(X_{k}|Z_{1:k-1})dX_{k}$$

Deploying Equations (2) to (5), the $P_{k|k}^{\left(i+1\right)}$ after iterative operations can be obtained as follows:(39)$$K_{k}^{\left(i+1\right)}=P_{k|k-1}^{\left(i+1\right)}H_{k}^{T}{\left(H_{k}P_{k|k-1}^{\left(i+1\right)}H_{k}^{T}+R_{k}^{\left(i+1\right)}\right)}^{-1}$$
(40)$$P_{k|k}^{\left(i+1\right)}=(I-K_{k}H_{k})P_{k|k-1}^{\left(i+1\right)}{\left(I-K_{k}H_{k}\right)}^{T}+K_{k}R_{k}^{\left(i+1\right)}K_{k}^{T}$$

After all, the PECM Λ can be obtained from $P_{k|k}^{\left(i+1\right)}$.

To achieve timely adjustment of the posterior approximation, a fading memory index weighting is used in this paper: (41)$$\sum_{i=1}^{k}\beta_{i}=1,\beta_{i}=\beta_{i-1}b,(0<b<1)$$
(42)$$\left\{\begin{array}{c} d_{k}=\frac{1-b}{1-b^{k}}\\ \beta_{k}=d_{k}b^{i-1} \end{array}\right.$$
where *b* is the forgetting factor, usually takes a value of between 0.95 and 0.99. The previous approximate posteriors are spread through the fading memory index weighting, and the prior parameters of $q(P_{k|k-1})$, $q(R_{k})$ can be given as follows:(43)$$\left\{\begin{array}{c} y_{k|k-1}=(1-\beta_{k})(y_{k-1|k-1}-n-1)+n+1\\ Y_{k|k-1}=(1-\beta_{k})Y_{k-1|k-1} \end{array}\right.$$
(44)$$\left\{\begin{array}{c} u_{k|k-1}=(1-d_{k})(u_{k-1|k-1}-m-1)+m+1\\ U_{k|k-1}=(1-d_{k})U_{k-1|k-1} \end{array}\right.$$

After N iterations, the VB approximations of the expected posterior PDFs are obtained:(45)$$\left\{\begin{array}{c} q(P_{k|k-1})\approx q^{\left(N\right)}(P_{k|k-1})=IW(P_{k|k-1};y_{k|k-1}^{\left(N\right)},Y_{k|k-1}^{\left(N\right)})\\ q(R_{k})\approx q^{\left(N\right)}(R_{k})=IW(R_{k};u_{k}^{\left(N\right)},U_{k}^{\left(N\right)})\\ q(X_{k})\approx q^{\left(N\right)}(X_{k})=N(X_{k};{\hat{X}}_{k|k}^{\left(N\right)},P_{k|k}^{\left(N\right)}) \end{array}\right.$$

### 3.2. Three-Dimensional ANP Evaluation Model

The traditional ANP evaluation model mainly focuses on the aircraft navigation performance along the track. Based on the assumption of three-dimensional independence and uniformity, the traditional model can only perform approximate calculations, and the evaluation results are not accurate. A 3D ANP evaluation model is proposed to accurately calculate the ANP in both horizontal and vertical planes.

PECM $\Lambda_{3D}$ for civil aircrafts in 3D space can be expressed as follows:(46)$$\Lambda_{3D}=A\Lambda_{3Dm}A^{-1}$$
where:(47)$$\Lambda_{3Dm}=\left[\begin{array}{ccc} \sigma_{mx}^{2} & 0 & 0\\ 0 & \sigma_{my}^{2} & 0\\ 0 & 0 & \sigma_{mz}^{2} \end{array}\right]$$

The probability *P* that the real position falls into the three-dimensional normal distribution integral ellipsoid domain is as follows:(48)$$P=\underset{\Omega }{∭}\frac{1}{{\left(2\pi \right)}^{3/2}{\left(\det\Lambda_{3Dm}\right)}^{1/2}}\exp\left(-\frac{1}{2}E^{T}\Lambda_{3Dm}^{-1}E\right)$$

For a given probability, the ANP calculation process can be converted into the calculation of the three-axis length of the integral ellipsoid Ω.

The probability integration can be simplified to the following:(49)$$P=2[\varphi (w)-\frac{w}{{\left(2\pi \right)}^{1/2}}\exp(-\frac{w^{2}}{2})]-1$$
where $w=R/\sigma_{mx}$, and φ(w) is the standard normal distribution function. For a given allowable error ξ, the ANP can be obtained via iterative numerical integration.

The flowchart of the proposed adaptive VB-based 3D ANP evaluation method is shown in Figure 5. The main process includes (1) the time update; (2) the initialization of VB iterative parameters based on fading exponential factors; (3) the approximation of $P_{k|k-1},R_{k},X_{k}$, which was obtained through N-times VB iterations; (4) the 3D ANP calculation based on the PECM.

## 4. Experimental Setup

To verify the effectiveness of the algorithm proposed in this paper, the performance of the following algorithms is compared, including the traditional ANP evaluation method based on KF with a nominal $R_{k}$ and $Q_{k}$ noise matrix(KFNN), traditional ANP evaluation method based on KF with a true $R_{k}$ and $Q_{k}$ noise matrix(KFTN), ANP evaluation method based on VBKF for estimating only $R_{k}$ (VBKFR), and the proposed 3D ANP evaluation method based on the VB-PECM estimator for estimating $R_{k}$ and $Q_{k}$.

The integrated navigation system loosely adopts the IRS/GNSS position combination. The simulation experiments are carried out in the MATLAB environment. The standard deviation of the GNSS positioning error for each leg is set as follows: (1) 10 m (take off, landing), (2) 20 m (departure, climb, descent, approach), (3) 50 m (en-route). The bias of airborne accelerometers is set as 10^−4^ g, and the bias of gyros is set as 0.01°/h [20,21,22].

The RNP flight procedure from Shanghai Hongqiao International Airport (ZSSS) to Chengdu Shuangliu International Airport (ZUUU) is executed in the simulation. The aircraft’s flight trajectory is dynamic, taking full account of the aircraft’s maneuverability, comprising five phases: takeoff, climb, steady flight, turn, descent and landing. The initial position is 31.1967° N, 121.335° E at an altitude of 8 m; the initial heading angle is 175°; and the arrival airport is 30.58° N, 103.948° E at an altitude is 495 m. The simulation duration is 10,505 s. The fly track simulation output is shown in Figure 6.

## 5. Results

Figure 7 shows the comparison results of the position-estimated error of the FMS in the east, north, and vertical directions of the different algorithms.

To evaluate the position estimate accuracy of the proposed algorithm, the root mean square error (RMSE) of position error is chosen as an evaluation metric. The formula for calculating RMSE is as follows:(50)$$\mathrm{RMSE}=\sqrt{\frac{1}{N_{S}}\sum_{i=1}^{N_{S}}{\left({\hat{x}}_{i}-{\bar{x}}_{i}^{\mathrm{true}}\right)}^{2}}$$
where $N_{S}$ is the number of data points, $\hat{x}$ is the position estimated value of FMS, and ${\bar{x}}_{i}^{\mathrm{true}}$ is the position true value of FMS obtained from the simulated track generator.

Table 1 shows the position error RMSE of each algorithm in three directions.

Table 2 and Table 3 show the mean and standard deviation of the position error for each algorithm in three directions. 

As can be seen from the position error curve and the statistical table of position errors, the proposed VB-PECM method can significantly improve the precision of integrated navigation. The addition of forgetting factor makes the tracking performance of measurement noise better, reducing the influence of time-varying noise on the navigation results.

The comparison of the traditional ANP method with KFNN, the traditional ANP method with VB-PECM, and the proposed 3D ANP method with VB-PECM that was proposed in this paper is shown in Figure 8.

To compare the success rate and precision of each ANP evaluation method, the F-score and mean absolute percentage error (MAPE) statistics are introduced as the evaluation metrics.
(51)F−Score=(1+τ)Precision⋅Recallτ2⋅Precision+RecallPrecision=TPTP+FPRecall=TPTP+FN

The F-score represents the comprehensive efficiency of the ANP evaluation, *TP* is the number of correct evaluations, *FP* is the number of misevaluations, *FN* is the number of missed evaluations, and τ is the harmonization factor of precision and recall. When τ=1, the F-score can fully represent the accuracy of each ANP model. 

MAPE denotes the approximability of ANP to the true position error.
(52)$$MAPE=\frac{1}{n}\sum_{t=1}^{n}\left|\frac{E_{t}-A_{t}}{E_{t}}\right|$$
where *n* is the number of data points, $E_{t}$ is the true position error of FMS, and $A_{t}$ is the ANP results in each direction. 

As can be seen, because the traditional ANP with KFNN uses a nominal $R_{k}$ and $Q_{k}$ noise matrix, the ANP evaluation result is a smooth curve, which has a large error gap compared to the real navigation positioning error. The traditional ANP with VB-PECM and the proposed ANP with VB-PECM accurately perceived the time-varying unknown noise in the integrated navigation system due to the PECM estimator. The ANP model proposed in this paper further improved the accuracy of the ANP evaluation. Table 4 and Table 5 show the F-1score and MAPE of each ANP method in three directions. 

Table 6 and Table 7 show the mean and standard deviation of ANP for each algorithm in three directions.

As can be seen from the ANP curve and the statistical table of ANP, the proposed 3D ANP model with VB-PECM has the lowest MAPE and the highest F1-score. This means the results of the ANP algorithm proposed in this paper are closer to the real navigation error, and the 3D model proposed in this paper makes the ANP evaluation in three directions more accurate. The statistical results of the mean and variance of each algorithm also show that the proposed algorithm has a higher performance.

In order to verify the effectiveness of the proposed algorithm, Monte Carlo simulations were carried out under the RNP flight procedure from Beijing Capital Airport (ZBAA) to Shanghai Hongqiao International Airport (ZSSS). The initial position is 40.073333° N, 116.598333° E at an altitude of 30.4 m, the initial heading angle is 175°, and the end point is 31.196667° N, 121.335000° E at an altitude is 8 m. The standard deviation of GNSS positioning error for each leg is set to be the same as the previous simulation conditions. The F1-score and MAPE of each ANP method in three directions are shown in Table 8 and Table 9.

Based on the above simulation results, it can be seen that the proposed VB-PECM filter has stronger adaptability to time-varying unknown noise. The proposed VB-PECM filter not only improves the accuracy of positioning but also the estimation accuracy of the PECM. Combined with the 3D ANP model proposed in this paper, the ANP evaluation is more effective. The evaluation accuracy in the east direction is 60.7% higher than the traditional ANP with KFNN, and it is 38.8% higher than the traditional ANP with VB-PECM. In the north direction, it is 55.8% higher than the traditional ANP with KFNN and 39.7% higher than the traditional ANP with VB-PECM. In the vertical direction, it is 56.9% higher than the traditional ANP with KFNN and 31.7% higher than the traditional ANP with VB-PECM.

## 6. Conclusions

During RNP operation, it is important to monitor the actual navigation performance of FMS accurately. We aimed to address the problem that the traditional ANP calculation method has regarding nominal noise matrices, where the Kalman filter cannot be sensitive to time-varying unknown noise, which will lead to inaccurate evaluation results. 

Firstly, this paper proposed a PECM estimator based on the variational Bayesian method. Accurate estimation of the PECM is achieved via iterative approximation of the predicted error covariance matrix and the measurement noise matrix. Meanwhile, the addition of fading memory index weighting endows the VB-PECM with a better tracking performance for time-varying noise. On this basis, the 3D ANP calculation model is constructed to solve the problem of missing evaluation dimensions and inaccurate evaluation of the traditional model. The simulation results show that the proposed ANP calculation method in this paper can effectively improve the evaluation accuracy of ANP during RNP operation with time-varying unknown noise. Thus, the proposed algorithm has a certain theoretical reference value for ensuring the flight safety of civil aircrafts under RNP operation.

## Figures and Tables

**Figure 1 sensors-24-05093-f001:**
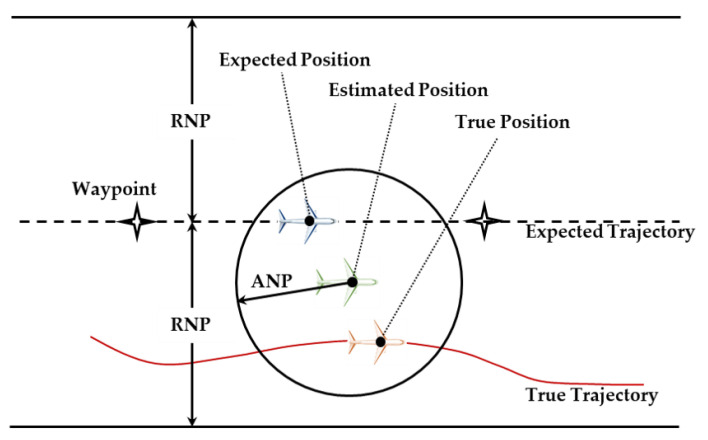
Schematic diagram of ANP definition.

**Figure 2 sensors-24-05093-f002:**
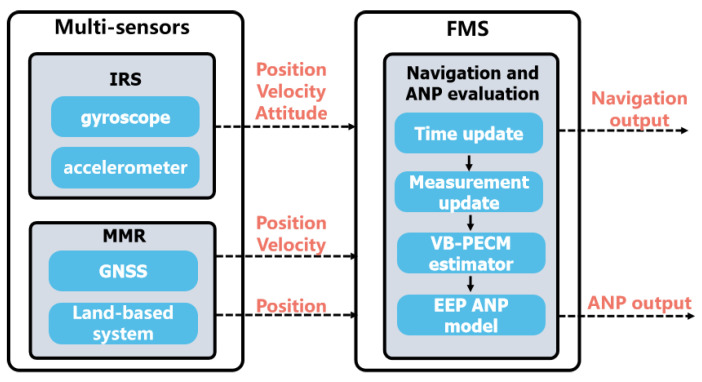
Proposed airborne integrated navigation and ANP evaluation system.

**Figure 3 sensors-24-05093-f003:**
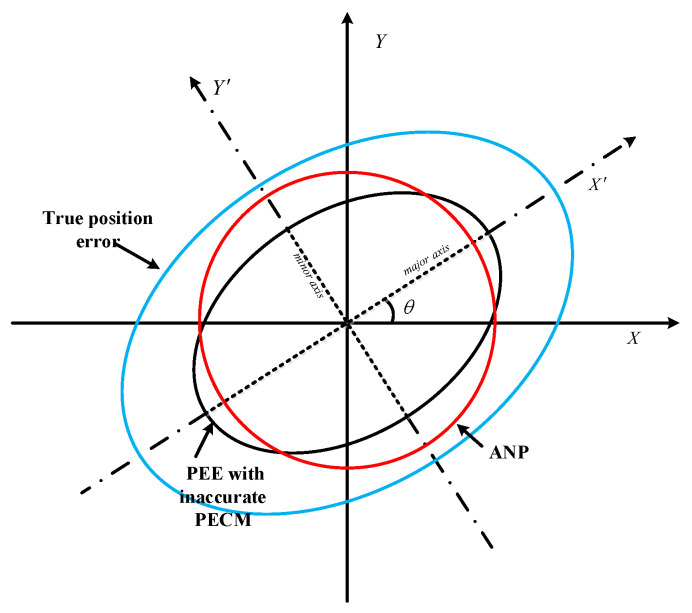
ANP misevaluation caused by inaccurate PECM.

**Figure 4 sensors-24-05093-f004:**
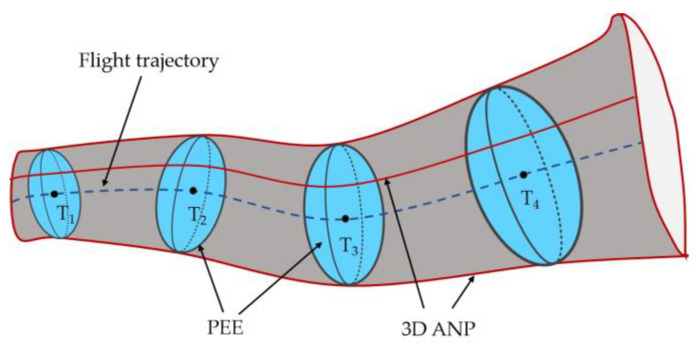
3D ANP tunnel during RNP flight.

**Figure 5 sensors-24-05093-f005:**
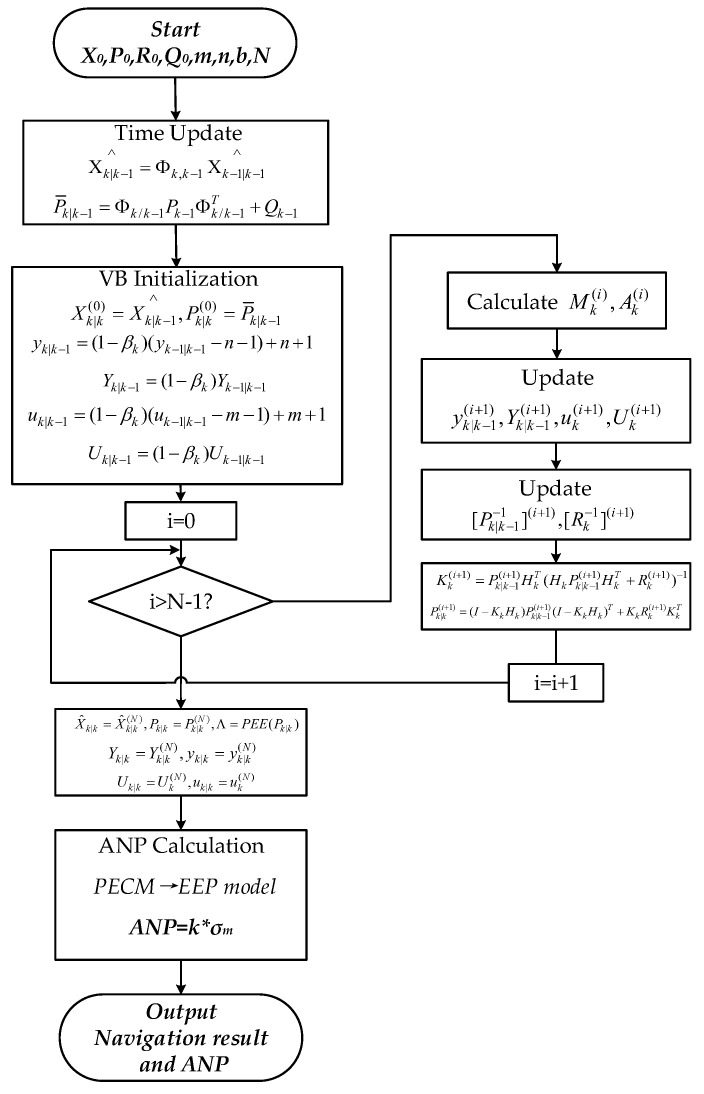
Three-dimensional ANP evaluation based on adaptive PECM estimation.

**Figure 6 sensors-24-05093-f006:**
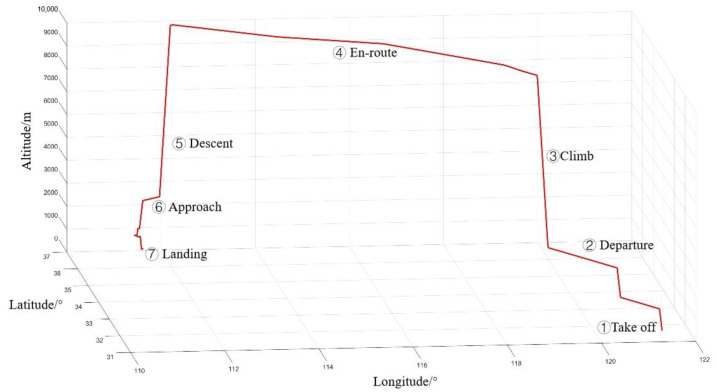
Fly track simulation output.

**Figure 7 sensors-24-05093-f007:**
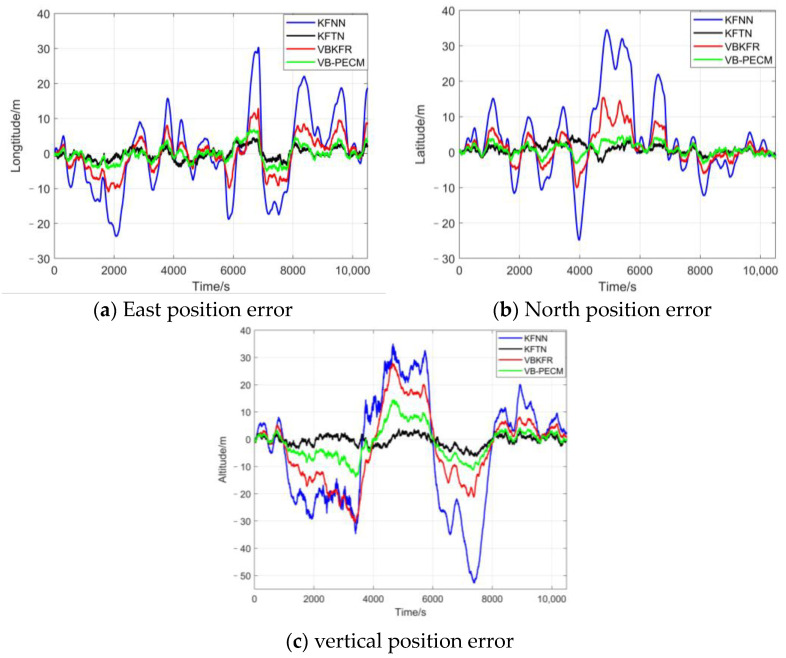
Comparison between navigation positioning errors.

**Figure 8 sensors-24-05093-f008:**
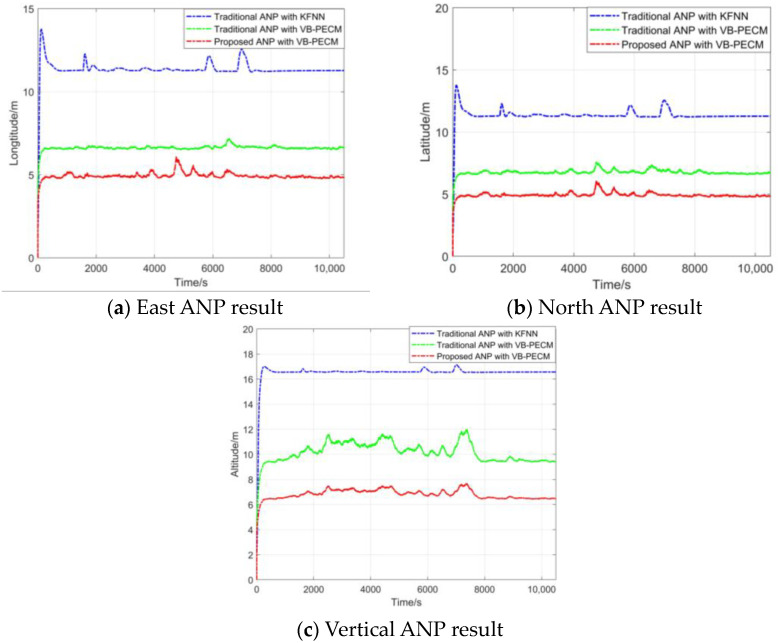
Comparison between different ANP evaluation results.

**Table 1 sensors-24-05093-t001:** Comparison of position error RMSE of each algorithm.

Position Error (RMSE)	East/m	North/m	Vertical/m
KFNN	11.20	11.95	21.52
KFTN	1.69	1.45	2.09
VBKFR	5.12	4.76	13.45
VB-PECM	2.53	1.83	5.98

**Table 2 sensors-24-05093-t002:** Comparison of position error mean of each algorithm.

Mean	East/m	North/m	Vertical/m
KFNN	−0.1	3.8	−4.45
KFTN	−0.18	0.73	−0.51
VBKFR	−0.33	1.53	−2.62
VB-PECM	−0.15	0.8	−1.11

**Table 3 sensors-24-05093-t003:** Comparison of position standard deviation of each algorithm.

Standard Deviation	East/m	North/m	Vertical/m
KFNN	11.2	11.3	21.05
KFTN	1.63	1.25	2.02
VBKFR	5.11	4.58	13.12
VB-PECM	2.53	1.65	5.87

**Table 4 sensors-24-05093-t004:** Comparison of F1-score of the results of each ANP algorithm.

F1-Score	East	North	Vertical
Traditional ANP with KFNN	80.3%	81.2%	66.3%
Traditional ANP with VB-PECM	96.5%	94.1%	89.8%
Proposed ANP with VB-PECM	99.7%	99.7%	98.9%

**Table 5 sensors-24-05093-t005:** Comparison of F1-score and MAPE of the results of each ANP algorithm.

MAPE	East/m	North/m	Vertical/m
Traditional ANP with KFNN	7.54	6.79	11.14
Traditional ANP with VB-PECM	4.57	4.99	7.13
Proposed ANP with VB-PECM	2.83	2.93	4.88

**Table 6 sensors-24-05093-t006:** Comparison of ANP mean of each algorithm.

Mean	East/m	North/m	Vertical/m
Traditional ANP with KFNN	11.36	11.36	16.51
Traditional ANP with VB-PECM	6.62	6.76	10.16
Proposed ANP with VB-PECM	4.92	4.85	6.18

**Table 7 sensors-24-05093-t007:** Comparison of ANP standard deviation of each algorithm.

Standard Deviation	East/m	North/m	Vertical/m
Traditional ANP with KFNN	0.6	0.6	0.87
Traditional ANP with VB-PECM	0.19	0.20	0.78
Proposed ANP with VB-PECM	0.15	0.15	0.38

**Table 8 sensors-24-05093-t008:** F1-score comparison of each ANP algorithm results in Monte Carlo simulations.

F1-Score	East	North	Vertical
Traditional ANP with KFNN	77.3%	75.1%	62.3%
Traditional ANP with VB-PECM	90.1%	89.7%	86.3%
Proposed ANP with VB-PECM	96.2%	96.5%	95.8%

**Table 9 sensors-24-05093-t009:** MAPE comparison of each ANP algorithm results in Monte Carlo simulations.

MAPE	East/m	North/m	Vertical/m
Traditional ANP with KFNN	8.34	7.12	12.37
Traditional ANP with VB-PECM	5.35	5.21	7.81
Proposed ANP with VB-PECM	3.27	3.14	5.33

## Data Availability

The original contributions presented in the study are included in the article, further inquiries can be directed to the corresponding author.

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
