# Peer review of "Adaptive Navigation Performance Evaluation Method for Civil Aircraft Navigation Systems with Unknown Time-Varying Sensor Noise"

_sensors, 2024, doi:10.3390/s24165093_

Round 1
Reviewer 1 Report
Comments and Suggestions for Authors
1.What the software conducted in this paper about experimental setup section?
2.What the contribution in this paper and should be highlight?
3.What the reason about in Fig.8, the performance in different ANP evaluation results have the same trend consistent but error value is very large?
Author Response
Dear Reviewer,
We sincerely appreciate the time and effort you both dedicated to reviewing our manuscript. Your feedback and comments have been extremely valuable in improving the quality of our work. Based on your suggestion and request, we have made corrected modifications on the revised manuscript. We hope that our work can be improved again. Furthermore, we would like to show the details as follows:
Comments 1: What the software conducted in this paper about experimental setup section?
Response 1: We appreciate it very much for this good suggestion, the simulation experiment was carried out in MATLAB environment, we made a supplementary explanation in the experimental section of the paper.
Revised: (Line 200-202)The integrated navigation system adopts IRS/GNSS position loosely combination, the simulation experiment was carried out in MATLAB environment, the standard deviation of GNSS positioning error for each leg is set as
Comments 2: What the contribution in this paper and should be highlight?
Response 2: Thank you for your suggestion, we optimized the contribution point description in the introduction section.
Revised: (Line 71-81) The main contribution of this paper are: 1) A novel VB-based PECM estimator was constructed to accurately estimate both the predicted error covariance matrix and the measurement noise matrix in integrated navigation system. This approach effectively addresses the issue of inaccurate estimation caused by unknown noise in the integrated navigation system input. 2) The fading memory index weighting is used to improve the tracking capability of time-varying noise, and the accuracy of the actual navigation performance of the integrated navigation system is further improved. 3) To address the limitations of the traditional ANP model, a 3D ANP model was developed, specifically designed to overcome issues such as missing evaluation dimension and model inaccuracy. Finally, a simulation has been given to prove the algorithm.
Comments 3: What the reason about in Fig.8, the performance in different ANP evaluation results have the same trend consistent but error value is very large?
Response 3: Thank you for your suggestion, the traditional ANP with KFNN uses nominal noise matrix, the ANP evaluation result is a smooth curve, but which has a large error gap with the real navigation positioning error. The proposed ANP method accurately perceived the time-varying unknown noise in the integrated navigation system. Therefore, the curve is smooth and consistent with VB-PECM results. We also made a supplementary explanation in the paper.
Revised:(Line 249-254) As can be seen that because the traditional ANP with KFNN uses nominal noise matrix, the ANP evaluation result is a smooth curve, which has a large error gap with the real navigation positioning error. The traditional ANP with VB-PECM and proposed ANP with VB-PECM accurately perceived the time-varying unknown noise in the integrated navigation system due to the PECM estimater, the ANP model proposed in this paper further improved the accuracy of the ANP evaluation.
Thank you for bringing this to our attention, and we appreciate your feedback.
Best regards
Reviewer 2 Report
Comments and Suggestions for Authors
(1) In the Introduction section, it is recommended to add updated references from 2023 and 2024 to enrich the introduction, here are some recommended literatures of recent years connected with sensors.
Integrating dynamic event-triggered and sensor-tolerant control: Application to USV-UAVs cooperative formation system for maritime parallel search, IEEE Transactions on Intelligent Transportation Systems.
Fault detection and fault-tolerant control for discrete-time multi-agent systems with sensor faults: A data-driven method, IEEE Sensors Journal
(2) The main contribution of this work should be listed in the introduction part of this paper.
(3) English structure and typo should carefully be checked and corrected.
(4) In Line 95, please describe the difference between ‘Pk|k’ and ‘Pk|k-1’ in the manuscript.
(5) Some pictures are distorted, please check carefully. E.g. Figure 8.
(6) In Section IV, how do you concern the RNP of the landing phase?
Based on above considerations, the current version is satisfied to be published in the Journal of Sensors. Thus, this paper is suggested minor revised to improve the current version.
Comments on the Quality of English LanguageEnglish structure and typo should carefully be checked and corrected. E.g. On Page 13, Line 247 "Table 6 and Table 7 shows " should be revised as "Table 6 and Table 7 show"
What's more, the formatting of citations should be double-checked and revised.
Author Response
Dear Reviewer,
We sincerely appreciate the time and effort you both dedicated to reviewing our manuscript. Your feedback and comments have been extremely valuable in improving the quality of our work. Based on your suggestion and request, we have made corrected modifications on the revised manuscript. We hope that our work can be improved again. Furthermore, we would like to show the details as follows:
Comments 1: In the Introduction section, it is recommended to add updated references from 2023 and 2024 to enrich the introduction, here are some recommended literatures of recent years connected with sensors.
Integrating dynamic event-triggered and sensor-tolerant control: Application to USV-UAVs cooperative formation system for maritime parallel search, IEEE Transactions on Intelligent Transportation Systems.
Fault detection and fault-tolerant control for discrete-time multi-agent systems with sensor faults: A data-driven method, IEEE Sensors Journal
Response 1: Thank you for your suggestion, We have added some references from recent years to strengthen the introduction section, including papers recommended by the reviewer.
Revised:(Line 35- 41) The flight management system (FMS) requires fault detection, fault isolation and system reconfiguration capabilities, FMS uses hardware redundancy and model-based analytic redundancy methods for accurate fault identification.[5-7] The FMS also needs to accurately evaluate civil aircraft’s navigation capability in real-time to determine if it meets the current RNP threshold. An incorrect evaluation result will cause misjudgement by the crew, which will lead to serious flight safety.
Comments 2: The main contribution of this work should be listed in the introduction part of this
Response 2: Thank you for your suggestion, we optimized the contribution point description in the introduction section.
Revised: (Line 71-81) The main contribution of this paper are: 1) A novel VB-based PECM estimator was constructed to accurately estimate both the predicted error covariance matrix and the measurement noise matrix in integrated navigation system. This approach effectively addresses the issue of inaccurate estimation caused by unknown noise in the integrated navigation system input. 2) The fading memory index weighting is used to improve the tracking capability of time-varying noise, and the accuracy of the actual navigation performance of the integrated navigation system is further improved. 3) To address the limitations of the traditional ANP model, a 3D ANP model was developed, specifically designed to overcome issues such as missing evaluation dimension and model inaccuracy. Finally, a simulation has been given to prove the algorithm.
Comments 3: English structure and typo should carefully be checked and corrected.
Response 3: We appreciate it very much for this good suggestion, we have proofread the full text for English grammar and expression.
Comments 4: In Line 95, please describe the difference between ‘Pk|k’ and ‘Pk|k-1’ in the manuscript.
Response 4: We appreciate it very much for this good suggestion, we added the relevant explanation.
Revised: (Line 103-104) Pk|k is the estimated error matrix (EEM) of the Kalman filter, which includes the position estimated error (PEE) of airborne integrated navigation system, Pk|k-1is the One-step prediction error matrix.
Comments 5:Some pictures are distorted, please check carefully. E.g. Figure 8.
Response 5: We appreciate it very much for this good suggestion, we have checked the full text and replaced the distorted images
Comments 6: In Section IV, how do you concern the RNP of the landing phase?
Response 6: We appreciate it very much for this good suggestion, the RNP for the landing phase follows reference 8 and selects the high-precision approach procedure RNP AR 0.1 from PBN as the RNP50/1000 real-time translation for the landing phase. The RNP for the landing phase follows reference 8 and selects the high-precision approach procedure RNP AR 0.1 from PBN as the RNP for the landing phase.
Thank you for bringing this to our attention, and we appreciate your feedback.
Best regards
Reviewer 3 Report
Comments and Suggestions for Authors
This paper deals with the navigation performance assessment, mainly about the position error estimation. I have several questions below:
1. What is the difference between the traditional filter-based position estimation and the studied actual navigation performance assessment? It seems no difference. If so, the topic should focus on the method of the filter-based estimation method. This is a well-studied field with fruitful results. Then, what is the main contribution of this paper. This should be clarified.
2. For figure 7, how to get the actual position information?
3. The experiment is not enough. Only one single case is considered.
Comments on the Quality of English Language
N/A
Author Response
Dear Reviewer,
We sincerely appreciate the time and effort you both dedicated to reviewing our manuscript. Your feedback and comments have been extremely valuable in improving the quality of our work. Based on your suggestion and request, we have made corrected modifications on the revised manuscript. We hope that our work can be improved again. Furthermore, we would like to show the details as follows:
Comments 1: What is the difference between the traditional filter-based position estimation and the studied actual navigation performance assessment? It seems no difference. If so, the topic should focus on the method of the filter-based estimation method. This is a well-studied field with fruitful results. Then, what is the main contribution of this paper. This should be clarified.
Response 1: Thank you for your suggestion. In the introduction section, we optimized the description of the contribution points of this paper, highlighting the contribution to the optimization of ANP evaluation methods.
Revised: (Line 71-81) The main contribution of this paper are: 1) A novel VB-based PECM estimator was constructed to accurately estimate both the predicted error covariance matrix and the measurement noise matrix in integrated navigation system. This approach effectively addresses the issue of inaccurate estimation caused by unknown noise in the integrated navigation system input. 2) The fading memory index weighting is used to improve the tracking capability of time-varying noise, and the accuracy of the actual navigation performance of the integrated navigation system is further improved. 3) To address the limitations of the traditional ANP model, a 3D ANP model was developed, specifically designed to overcome issues such as missing evaluation dimension and model inaccuracy. Finally, a simulation has been given to prove the algorithm.
Comments 2: For figure 7, how to get the actual position information?
Response 2: We appreciate it very much for this good suggestion, the position true value of FMS obtained from the simulated track generator, we made a supplementary explanation in the paper.
Revised: (Line221-222) the position true value of FMS obtained from the simulated track generator.
Comments 3:The experiment is not enough. Only one single case is considered.
Response 3: We appreciate it very much for this good suggestion, we consider different experimental scenarios and supplement Monte Carlo simulation experiments under other flight paths to ensure the adequacy of the algorithm verification in this paper.
Revised:(Line270-288) In order to verify the effectiveness of the proposed algorithm, Monte Carlo simulations were carried out under the RNP flight procedure from Beijing Capital Airport (ZBAA) to Shanghai Hongqiao International Airport (ZSSS), the initial position is 40.073333° N, 116.598333° E at an altitude of 30.4 m, the initial heading angle is 175°, and the end point is 31.196667°N, 121.335000°E at an altitude is 8 m. The standard deviation of GNSS positioning error for each leg is set as same as previous simulation conditions, the F-1score and MAPE of each ANP method in three directions were shown in Table 8 and Table 9.
Thank you for bringing this to our attention, and we appreciate your feedback.
Best regards
Round 2
Reviewer 2 Report
Comments and Suggestions for Authors
The comments are addressed.
Reviewer 3 Report
Comments and Suggestions for Authors
My concerns are answered